# Theoretical and Experimental Study of Changes in the Structure of the Intermediate Layer during Friction between Contacting Bodies

**DOI:** 10.3390/ma14195689

**Published:** 2021-09-30

**Authors:** Jeng-Haur Horng, Nikolay M. Osipenko, Fedor I. Stepanov, Elena V. Torskaya

**Affiliations:** 1Power Mechanical Engineering Department, National Formosa University, Yunlin 632, Taiwan; jhhorng@gmail.com; 2Ishlinsky Institute for Problems in Mechanics, Russian Academy of Sciences, 119526 Moscow, Russia; osipnm@mail.ru (N.M.O.); stepanov_ipm@mail.ru (F.I.S.)

**Keywords:** contact, friction, fracture, third body, BEM, wear particles

## Abstract

Friction is often accompanied by local fracture at the boundary of contacting bodies. The space between contacting bodies usually contains moving particles of a different nature, and a change in the effective friction conditions can be associated with a change in the structure of the contact area. This paper presents a new series of experiments where balls simulated the particles of the intermediate layer interacting with an elastic layer of different thickness. The effects of regularization when the balls approached each other were investigated considering different initial configurations (line and spatial structure). The balls simulated the particles of the intermediate layer interacting with the elastic layer of different thickness. The opposite effects of convergence and separation of the balls were observed in different experiments. A model of mutual effect during the contact of two balls with a two-layered elastic half-space was developed. An analysis of tangential forces due to the mutual effect was performed for different layer thicknesses, its relative compliance, and different distances between the balls. It was found that the input parameters defined the sign of the tangential force, which led to the convergence or the separation of the balls. The results can be used to create structures controlling the motion in the intermediate layer.

## 1. Introduction

During friction, wear particles are usually separated from interacting bodies. These particles, as well as particles coming from the outside, form a third body that changes the conditions of frictional interaction. An intermediate layer (third body) is formed, and further interaction of the bodies is carried out through it. The validity of such formulation of this situation is confirmed by numerous experimental data [1,2]. Systematic fracture in the presence of the third body at the contact boundary of two interacting bodies during shear can be one of the main wear mechanisms. The stage of formation of an intermediate layer was identified [3,4], and a variant of a flat element evolution of the third body was analyzed in the mode of intense tangential action on its contact surfaces. A model was developed to analyze the process leading to the appearance of elements of the third body during the destruction of material in the area of dry frictional contact.

The evolution of the intermediate layer and its effect on the mechanisms of friction and wear depend on the nature of the material and on the contact conditions. The separated fragment can form a rolling element or behave as a flat sliding element. This depends on the ratio of normal pressure and shear in the contact area. A criterion for the initiation of transformation of a flat element into a rolling element was described [5].

An interaction between bodies in an area of intense friction occurs through the movement (rolling) of structural elements of the intermediate layer, which include both wear particles and those appearing from the outside. The latter can include, for example, abrasive particles. The influence of interface particles with different hardness and plasticity index on the contacting bodies is an important factor affecting the interface temperature [6]. Moreover, in the analysis of experiments on ball screws, it was also found that the particle size and concentration have a significant influence on wear and contact temperature [7]. The rolling ratio of the particles can reduce the friction coefficient, which is also one of the factors affecting friction [8]. However, the relation between the structure of the intermediate layer and friction has not been extensively discussed.

Contact interaction in the presence of friction and wear is characterized by self-regulation mechanisms, which result in the formation of a regular structure corresponding to an intermediate layer. In [9,10], one of the mechanisms of such self-regulation was presented; it is based on the possibility of a relative rearrangement of particles in the intermediate layer under the action of small perturbations of the stress state in their vicinity. In a steady regime, the preference of various states of the friction structures is associated with the value of the total potential energy of the elastic system corresponding to these states. The state of the system with less total potential energy is more probable. In this case, the nature of bodies interaction at the macro level depends on the mutual position of the elements in the intermediate layer. The mechanism regulating the mutual position of the intermediate layer particles is the subject of this study.

The following example illustrates the proposed approach. Let identical solid spherical particles located at a small distance from each other be pressed into an elastic half-space under the effect of normal loads Pi. Due to the finiteness of the distance between the particles, small forces $T_{i}$, directed towards each other, appear, which is typical for closely spaced dies. During static loading, this does not lead to displacement of the dies. The situation changes if the balls are set in motion at a certain velocity along the contact surface (for example, by moving a rigid plate that transfers a load to them). In this case, it is possible to obtain the phenomenon of transformed friction [11]. According to it, if the body moves under the action of a force in the friction mode, then the appearance of an additional force in another direction, in particular, the force $T_{i}$, leads to the displacement of the body in the direction of action of this force, no matter how small it is. For this mode of motion, there is no fundamental difference between sliding and rolling friction. Such interaction provides a mechanism that can change the mutual position of elements in the intermediate layer between the contacting bodies during friction. It was shown [11] that in the case when the initial elements of the intermediate layer are spherical blocks capable of rolling under the effect of tangential forces, normal loads from the side of the contacting bodies lead to the convergence of the blocks as the total tangential displacements increase. This model approach was confirmed experimentally [9,10]. It can be extended to a system of blocks moving relatively to each other in the sliding mode. The rate of convergence of the blocks does not depend on the direction of the main displacement of the contacting bodies. The described process is the reason for the grouping of particles in the intermediate layer and their subsequent arrangement into some compact formations.

This paper presents a new series of experiments to study regularization effects for various initial configurations of a system of balls (line and spatial structure). The balls imitate the particles of the intermediate layer interacting with an elastic body. Previously, the motion and the effect of convergence in the intermediate layer were associated with the reaction of a homogeneous elastic half-space. The presence of an additional layer between the rigid base and the third body can change the scenario of the structure’s development. In the case of a thin elastic interlayer on a rigid foundation, scattering of initially closed particles becomes probable. To explain this effect, it is necessary to use models of the interaction of particles with a layered base, in which the mutual effect is taken into account. Methods and approaches used in this study were developed to solve the 3D contact problem for coated solids [12]. The mutual effect was studied previously for a periodic system of dies and a two-layer elastic foundation [13]. Here, the asymmetry effects arising from the mutual influence of two balls in contact with a two-layer elastic half-space are investigated.

In this study, only the mechanical component of the interaction of the particles of the third body is investigated. The effect of energy loss on the collision of particles in their position close to each other is not considered. This effect leads to increased friction losses. At the microscale there is also adhesion of the particle to the contact surfaces [14]; when approaching, molecular interaction between the particles also occurs [15], which additionally increases the dissipation of energy during friction. Probably, the assembly or dispersal of particles should affect tribochemical processes, for example, oxidation [16]. The real third body consists of microparticles of various sizes; the particles can be of different nature and shape and have different mechanical properties. The physical and mathematical models presented here make it possible to isolate the effect of mutual influence from a large number of factors influencing the behavior of the particles of the third body.

## 2. Physical Modeling of Structure Changes in the Third Body

An experimental method was developed to demonstrate the effects of motion in the intermediate layer. Steel balls were placed in a fixed gap between an elastic compliant layer and a rigid movable plate (Figure 1). The plate, made of clear glass with a thickness of 10 mm, received a reciprocating motion with amplitude of 7–10 cm. Observations of the relative displacement of the balls were made and photographed through the glass. All control experiments were carried out with a fixed vertical approach of the upper plate and the substrate. The displacement was controlled by fixing the level of the plate while sliding along the sides of the container (Figure 1).

### 2.1. Thick Layer

Identical steel balls of 10 mm radius and a compliant foam rubber layer 80 mm thick, were used in the first part of the experiments. The effective modulus of elasticity of the rubber, determined from the force during the test penetration of the ball to a depth of 12 mm, was *E* = 57.4 kPa. The fixed vertical approach of the upper plate to the substrate was 12 mm.

Four balls in line first were divided into groups, starting from the edges of a straight chain (line) of balls, in which this process was accelerated (Figure 2a). With an increase in the initial distance between the balls, the effect of dividing into groups, starting from the edges of the line, was more evident. Figure 2b shows the change in the distance between the centers of the balls for the variant of the line of four balls with the initial distance between them of 30 mm. Such type of grouping is caused by the ‘edge effect’: it means that, having the same penetration, the edge balls are loaded more than the central ones [1]. Two groups formed and then interacted with each other. The balls in the photos are numbered to track their mutual movements.

We then considered some examples of grouping lines with a larger number of balls.

For a line of six balls (Figure 3), we could see a sequential formation of two groups of balls—the outer ones grouped together and then one ball from the central group joined them (that is, two groups of three balls formed). Here, the process also began from the edge balls (30 cycles). The process ended with the formation of a compact group with the closest possible packing.

An interesting factor to analyze was the influence of irregularities in the line structure (Figure 4). A gap was created in the regular initial line structure of seven balls (one ball was outside the line). There was a sequential formation of two groups of balls–the grouping of the edge ones, the joining to them of one ball from the central ones, and then their merging, but the shape of the line curved towards the influencing separate ball. We also observed a gradual moving of an individual ball to the curved line.

We also carried out experiments with other configurations of the groups of balls. Figure 5 and Figure 6 present the results for nine balls placed at different initial distances from each other. The convergence was more active along the direction of the main movement. The final shape of the group showed a “tip” oriented towards the direction of the main movement.

It can be noted that with a small (20 mm) distance between the balls in the initial ordered configuration, the balls grouped together preserving elements of the original ordering; with an increase in the distance (up to 30 mm), the final configuration with the maximum density was the same, but in intermediate situations, more complex structures formed (possibly associated with an increased influence of random factors with a decrease in the initial density). It is possible that in the case of a much larger number of elements, even with a regular structure, several centers of grouping can be formed in the system.

### 2.2. Thin Layer. Effect of Balls Separation

Experiments were carried out with balls of various diameters (7 mm, 10 mm, 16 mm) moving along a 10 mm-thick layer of spongy rubber on a rigid substrate. A glass plate performed a reciprocating motion with an amplitude of 30–40 mm. The plate normal displacement was controlled to be 1.5 mm. When moving from the initial closed position, balls with a diameter of 16 mm were displaced to a distance of about 50 mm from the centers after more than 50 cycles of movements (Figure 7). This mutual location was final; it did not change with further reciprocated sliding. Close balls of smaller diameter were not separated. This and other effects are explained in the next section, which presents a model for the emergence of tangential forces due to the mutual effect.

## 3. Theoretical Model of Mutual Effect

The study of mutual effect when investigating multiple contacts is usually associated with surface roughness. For the case of elastic half-space, a periodic system of dies as well as a fixed number of dies were considered [1]. The mutual effect leads to a not uniform force distribution between a limited assembly of dies with controlled penetration, as well as to the increase of the tangential force because of a non-symmetric pressure distribution. The tangential force was analyzed in the case of a single slider in contact with elastic [17] and viscoelastic [18] solids. In the presence of asperities, such tangential force cannot produce any significant effect. The contact problem for the system of balls is mathematically identical to the problem for the system of dies, but the balls can change their location due to the tangential force.

The presented results of physical modeling should be explained by an appropriate mathematical model. The effect of balls convergence or separation in the reciprocated sliding process depends on the thickness of the compliant material layer, the balls’ size, and the normal force. These parameters, as well as the elastic properties of the materials in contact with each other were the input variables of the model.

Four important notes:

The model was developed for the case of small deformations, which means that the characteristic size of the contact area of a ball was assumed to be much smaller than the ball radius. For this case, the model is semi-analytical and applicable for the description of such effect as the appearance of tangential forces due to mutual influence. For the case of finite deformations, the effect should be stronger.

We assumed that it was important to use a low-compressible material for the layer (if we wanted the balls to separate). Experiments were performed with rubber with a large Poisson ratio. The model presented below can be used for all elastic materials, but the results were obtained for a Poisson ratio (ν) of ν=0.48. This value is typical for rubbers.

We ignored the rheological properties of the rubber, as well as the adhesion between the rubber and the balls. This last parameter is very important at the microscale for wear of abrasive particles; it should have a significant influence but will not neutralize the effects arising from elastic deformation.

Here, we considered only the loading of balls by normal forces. Tangential forces, which occur due to the mutual effect, lead to the differences in the sliding contact of neighboring balls.

### 3.1. Problem Formulation

Let us consider a contact problem for n rigid indenters whose shapes are defined by the functions $f_{i}(x,y),\;i=1..n$ and a two-layered elastic half-space. The layer has a thickness h. The points of initial contact of the indenters with the layer surface are located arbitrarily. The elastic layer of thickness h is bonded to an elastic half-space. The elastic properties of the materials are characterized by the Young modulus and the Poisson ratio $E_{1,\;2},\;\nu_{1,\;2}$. The indexes “1” and “2” correspond to the layer and the half-space, respectively. The Oz axis is directed normally to the undeformed surface of the layer.

Two different contact problems can be considered:

Problem 1—the indenters maintain their relative position in all directions, and the vertical force Q affects the system of the indenters as a unified object;

Problem 2—the indenters maintain their relative position within the plane XY but have a degree of freedom in the Oz axis. Herewith, the vertical forces $Q_{i},i=1..n$ acting on the indenters are given individually.

For Problem 1, the following boundary and equilibrium conditions are met (z=0):(1)$$\begin{array}{l} \sum_{q=1}^{n}w_{iq}(x,y)=f_{i}(x,y)+D\;,\;(x,y)\in \Omega_{i},\;i=1..n,\\ \sigma_{z}=0,\;(x,y)\notin \Omega_{i}\\ \tau_{xz}=0,\;\tau_{yz}=0\; \end{array},$$
(2)$$Q=\sum_{i=1}^{n}\iint_{\Omega_{i}}p(x,y)dxdy,\;i=1..n..$$

For Problem 2, the following boundary and equilibrium conditions are met (z=0):(3)$$\begin{array}{l} \sum_{q=1}^{n}w_{iq}(x,y)=f_{i}(x,y)+D_{i},\;(x,y)\in \Omega_{i},\;i=1..n\\ \sigma_{z}=0,\;(x,y)\notin \Omega_{i}\\ \tau_{xz}=0,\;\tau_{yz}=0\; \end{array},$$
(4)$$Q_{i}=\iint_{\Omega_{i}}p(x,y)dxdy,\;i=1..n,.$$

Here, $\Omega_{i},\;i=1..n$ are unknown contact zones of the indenters; the functions $w_{iq}(x,y)$ define surface displacement within the contact area of indenter *i* due to impact of indenter *q*; $D_{i},\;i=1..n$—are the vertical displacements of the indenters; $\sigma_{z}$, $\tau_{xz}$, $\tau_{yz}$ are normal and tangential stresses. Contact pressure and contact areas have to be determined. 

The conditions at the layer–substrate interface (z=−h) satisfied the case of perfect adhesion:(5)$$\begin{array}{l} w^{\left(1\right)}=w^{\left(2\right)},\;\;u_{x}^{\left(1\right)}=u_{x}^{\left(2\right)},\;\;u_{y}^{\left(1\right)}=u_{y}^{\left(2\right)},\\ \sigma_{z}^{\left(1\right)}=\sigma_{z}^{\left(2\right)},\;\;\tau_{xz}^{\left(1\right)}=\tau_{xz}^{\left(2\right)},\;\;\tau_{yz}^{\left(1\right)}=\tau_{yz}^{\left(2\right)}\;\;\; \end{array}$$

Here $u_{x}$, $u_{y}$ are tangential displacements.

### 3.2. Method of Solution

Due to the mutual effect, the contact problem for each indenter is not axisymmetric. The first step was the loading of the two-layered elastic half-space by a load q, which is uniformly distributed inside a square 2a×2a. The loading conditions at the upper layer bound are the following:(6)$$\begin{array}{l} \sigma_{z}=-q,\;|x|\leq a,\;|y|\leq a\\ \sigma_{z}=0,\;|x|>a,\;|y|>a\\ \tau_{xz}=0,\;\tau_{yx}=0\; \end{array}$$

The Equations (5)–(6) are solved [19] using methods based on double Fourier transforms. Stresses and displacements inside the layer and the half-space are obtained as a result of inverse Fourier transforms. The normal displacements of the surface are determined by the following relation:(7)$$w^{\prime }(x^{\prime },y^{\prime },0)=-\frac{1+\nu_{1}}{E_{1}}\int_{0}^{\pi /2}\int_{0}^{\infty }\Delta (\gamma ,\varphi ,\lambda ,\chi )\cos(x^{\prime }\gamma \cos\varphi )\cos(y^{\prime }\gamma \sin\varphi )d\gamma d\varphi$$

Here, $x^{\prime }$, $y^{\prime }$, $w^{\prime }$ are dimensionless coordinates related to a, χ is the ratio of the reduced moduli of elasticity of the layer and the elastic half-space γ, φ are the coordinates in the space of double Fourier transforms, λ=h/a is dimensionless layer thickness. The function Δ(γ,φ,λ,χ) is obtained as a result of solving a system of linear functional equations obtained from the boundary conditions (5), (6) as a result of using biharmonic functions to determine stresses and displacements, as well as a double integral Fourier transform applied to a constant load. The full analytical representation of Δ(γ,φ,λ,χ) is cumbersome and is not presented here, but it is important to note that this function linearly depends on $\bar{q}$, which is the result of applying the double Fourier transform to the constant pressure q:(8)$$\bar{q}=q\frac{4}{\pi^{2}}\frac{\sin(\gamma \cos\varphi )\sin(\gamma \sin\varphi )}{\gamma^{2}\sin\varphi \cos\varphi }$$

Due to the fact that the constant pressure q enters the function Δ(γ,φ,λ,χ) linearly and can be taken outside the integral sign, (7) can be taken as a basis for solving the contact problem by determining the contact pressure p(x,y) as a piecewise function. Thus, we use here the boundary element method.

A rectangle area $\Omega_{i}^{0},\;i=1..n$ of size $a_{i}\times b_{i}$ containing a priory the sought contact area is chosen for each indenter, taking into account its size, shape, and applied load. Every $\Omega_{i}^{0}$ contained an appropriate grid of $Na_{i}\cdot Nb_{i}=N_{i}$ elements. Contact pressure $p_{j}^{i},\;i=1..n,\;j=1..N_{i}$ is assumed constant within each element of the grid.

Vertical displacement at any point of the boundary is a superposition of displacements caused by all loaded elements. Consider a column $\left\|w_{}^{iq}\right\|$ to be composed of surface displacements within the area $\Omega_{i}^{0}$ due to the impact of pressure $\left\|p^{q}\right\|$ within the area $\Omega_{q}^{0}$. In this case, the relation between contact pressure and vertical displacements of the surface can be defined by the matrix of influence coefficients $\left\|A^{iq}\right\|$ consisting of $N_{i}\times N_{q}$ elements:(9)$$\left\|A^{iq}\right\|\cdot \left\|p^{q}\right\|=\left\|w^{iq}\right\|$$

Thus, the coefficients of the matrix $A_{jl}^{iq},\;j=1..N_{i},\;l=1..N_{q}$, which are the surface vertical displacements in the center of the element *j* of area $\Omega_{i}^{0}$ caused by a unit pressure within the element *l* of the area $\Omega_{q}^{0}$, can be calculated using the relationship (7).

Taking into account the introduced piecewise functions, one can rewrite the boundary and equilibrium conditions (1), (2) in a matrix form:(10)$$\left[\begin{array}{cccc} \left(\begin{array}{ccc} A_{11}^{11} & \cdots & A_{N_{1}1}^{11}\\ \vdots & \ddots & \vdots \\ A_{1N_{1}}^{11} & \cdots & A_{N_{1}N_{1}}^{11} \end{array}\right) & \cdots & \left(\begin{array}{ccc} A_{11}^{1n} & \cdots & A_{N_{n}1}^{1n}\\ \vdots & \ddots & \vdots \\ A_{1N_{1}}^{1n} & \cdots & A_{N_{n}N_{1}}^{1n} \end{array}\right) & 1\\ \vdots & \ddots & \vdots & \vdots \\ \left(\begin{array}{ccc} A_{11}^{1n} & \cdots & A_{N_{1}1}^{1n}\\ \vdots & \ddots & \vdots \\ A_{1N_{n}}^{1n} & \cdots & A_{N_{1}N_{n}}^{1n} \end{array}\right) & \cdots & \left(\begin{array}{ccc} A_{11}^{nn} & \cdots & A_{N_{n}1}^{nn}\\ \vdots & \ddots & \vdots \\ A_{1N_{n}}^{nn} & \cdots & A_{N_{n}N_{n}}^{nn} \end{array}\right) & 1\\ \left(\begin{array}{ccc} s^{1} & \cdots & s^{1} \end{array}\right) & \cdots & \left(\begin{array}{ccc} s^{n} & \cdots & s^{n} \end{array}\right) & 0 \end{array}\right]\cdot \left[\begin{array}{c} \left(\begin{array}{c} p_{1}^{1}\\ \vdots \\ p_{N_{1}}^{1} \end{array}\right)\\ \begin{array}{c} \vdots \\ \begin{array}{c} \left(\begin{array}{c} p_{1}^{n}\\ \vdots \\ p_{N_{n}}^{n} \end{array}\right)\\ D \end{array} \end{array} \end{array}\right]=\left[\begin{array}{c} \left(\begin{array}{c} f_{1}^{1}\\ \vdots \\ f_{N_{1}}^{1} \end{array}\right)\\ \begin{array}{c} \vdots \\ \begin{array}{c} \left(\begin{array}{c} f_{1}^{n}\\ \vdots \\ f_{N_{n}}^{n} \end{array}\right)\\ Q \end{array} \end{array} \end{array}\right]$$

Here, $s^{i},\;i=1..n$ are the areas of mesh elements, and $f_{j}^{i},\;i=1..n,\;j=1..N_{i}$ are piecewise functions defining the indenters’ shapes.

Conditions (3) and (4) thus can be rewritten in the following form:(11)$$\left[\begin{array}{cccc} \left(\begin{array}{ccc} A_{11}^{11} & \cdots & A_{N_{1}1}^{11}\\ \vdots & \ddots & \vdots \\ A_{1N_{1}}^{11} & \cdots & A_{N_{1}N_{1}}^{11} \end{array}\right) & \cdots & \left(\begin{array}{ccc} A_{11}^{1n} & \cdots & A_{N_{n}1}^{1n}\\ \vdots & \ddots & \vdots \\ A_{1N_{1}}^{1n} & \cdots & A_{N_{n}N_{1}}^{1n} \end{array}\right) & 1\\ \vdots & \ddots & \vdots & \vdots \\ \left(\begin{array}{ccc} A_{11}^{1n} & \cdots & A_{N_{1}1}^{1n}\\ \vdots & \ddots & \vdots \\ A_{1N_{n}}^{1n} & \cdots & A_{N_{1}N_{n}}^{1n} \end{array}\right) & \cdots & \left(\begin{array}{ccc} A_{11}^{nn} & \cdots & A_{N_{n}1}^{nn}\\ \vdots & \ddots & \vdots \\ A_{1N_{n}}^{nn} & \cdots & A_{N_{n}N_{n}}^{nn} \end{array}\right) & 1\\ \left(\begin{array}{ccc} s^{1} & \cdots & s^{1} \end{array}\right) & \cdots & 0 & 0\\ \vdots & \ddots & \vdots & \vdots \\ 0 & \cdots & \left(\begin{array}{ccc} s^{n} & \cdots & s^{n} \end{array}\right) & 0 \end{array}\right]\cdot \left[\begin{array}{c} \left(\begin{array}{c} p_{1}^{1}\\ \vdots \\ p_{N_{1}}^{1} \end{array}\right)\\ \vdots \\ \left(\begin{array}{c} p_{1}^{n}\\ \vdots \\ p_{N_{n}}^{n} \end{array}\right)\\ D_{1}\\ \vdots \\ D_{n} \end{array}\right]=\left[\begin{array}{c} \left(\begin{array}{c} f_{1}^{1}\\ \vdots \\ f_{N_{1}}^{1} \end{array}\right)\\ \vdots \\ \left(\begin{array}{c} f_{1}^{n}\\ \vdots \\ f_{N_{n}}^{n} \end{array}\right)\\ Q_{1}\\ \vdots \\ Q_{n} \end{array}\right]$$

The coefficients $A_{jl}^{iq},\;j=1..N_{i},\;l=1..N_{q}$ are obtained from (7):(12)$$A_{jl}^{iq}=-\frac{1}{2G}\int_{0}^{\pi /2}\int_{0}^{\infty }\Delta^{\prime }(\gamma ,\varphi ,\lambda )\cos(y_{jl}^{iq}\gamma \sin\varphi )\cos(x_{jl}^{iq}\gamma \cos\varphi )d\gamma d\varphi$$

Here, ${\left({\left(x_{jl}^{iq}\right)}^{2}+{\left(y_{jl}^{iq}\right)}^{2}\right)}^{1/2}$ is the distance between corresponding elements of the grids, $\Delta^{\prime }(\gamma ,\varphi ,\lambda )=\Delta (\gamma ,\varphi ,\lambda )/q$.

According to the boundary conditions (1), (3), the unknown contact pressure should be 0 outside the contact area. The systems (10) and (11) do not take into account this fact, so a solution will include negative pressure elements. In order to obtain the unknown contact areas, one should consider these elements to be zero and consequently reduce the system rank. After, the reduced system is solved again. The described iteration procedure continues until there are no negative pressure elements in the solution. As a result, one obtains the unknown contact areas, contact pressure, and normal displacement of the indenters.

The mutual effect leads to a non-axisymmetric contact pressure distribution under each indenter. It means that we have tangential forces with opposite directions, which arise for two balls. It is defined by the following relation:(13)$$\vec{T_{i}}=-\iint_{\Omega_{i}}{\vec{p}}_{XY}(x,y)dxdy,\;i=1..n$$

Here, ${\vec{p}}_{XY}(x,y)$ is a projection of the pressure on the plane *XY*. It can be calculated for an obtained solution as the surface displacement, and therefore, the surface curvature can be found.

### 3.3. Results and Discussion

First let us consider the simplest example, which can explain the nature of forces of attraction or repulsion during contact of the balls with a low compressible layer adherent to a more rigid substrate. What are the surface normal displacements if a two-layered elastic half-space is subjected to a load q, which is uniformly distributed inside a square 2a×2a? For an elastic half-space, the displacements are always positive (if the acting force is positive). In contrast, for the layer, the displacements are positive inside the loaded area and in the vicinity of the area. As we move away from the loading region, the displacements become negative, reach a certain minimum value $w^{*}$ at a point $x^{*}$, and then asymptotically tend to zero. Figure 8 presents the dependences of $w^{*}$ and $x^{*}$ on the thickness of a compliant low compressible layer. If the layer has zero thickness, the elastic half-space presents no negative displacements of the surface. For a thin layer, the negative displacements are located very close to the loaded area. The thicker the layer, the farther from the loading zone is the region of greatest bulging. The value of $w^{*}$ depends on the layer thickness non-monotonically. With an increase in the layer thickness, its absolute value increases and then decreases. It is clear that if the bulge is between two balls, they will separate. However, the opposite case is also possible, if both balls are in the region of positive displacements. In order to understand which scenario will be realized, it is necessary to solve the contact problem for the system of balls.

The problem formulated in Section 3.1 can be used for the case of a limited number of smooth indenters with various shapes. The calculations were performed for balls with radius R. We chose a two-ball configuration for the analysis of tangential forces due to mutual influence, since the separation was experimentally obtained for this configuration. In addition, having two equally loaded balls, we could reduce the number of input parameters and facilitate the analysis. The points of initial contact of the balls with the layer surface were located on the axis Ox; the distance between the balls was L. For a more generalizable analysis, we used the following dimensionless parameters: $(x^{\prime },\;y^{\prime },\;z^{\prime })=\left(x,\;y,\;z\right)/R$, $P^{\prime }=P/E_{1}$, $Q^{\prime }=Q/\left(E_{1}\cdot R^{2}\right)$, $L^{\prime }=L/R$.

The asymmetry of the contact pressure distribution is illustrated in Figure 9. The two contact pressures were close to each other (two curves appear like one in Figure 9a), but the difference between the two pressure distributions is obvious. This suggests that we tangential forces should be acting in opposite directions.

Let us consider the tangential force acting on the left ball. Since the axis was directed from left to right, positive forces brought the balls closer and negative forces separated the balls. For the ball on the right, this was exactly the opposite. The smaller the distance between the balls, the more noticeable the mutual effect. First, we analyzed the influence of the layer thickness when the balls were close to each other, $L^{\prime }=2$ (Figure 10). In this Figure, the starting point (zero thickness) corresponds to the case of an elastic half-space, when a very small positive force acts to bring together the balls. Next, we increased the thickness of the relatively compliant weakly compressible layer. Since the layer was thin and weakly compressible, some surface bulging of the material between the balls contact zones was inevitable. This resulted in a negative tangential force separating the balls. At a particular a layer thickness, this force reached its minimum. With a further increase in the layer thickness, a positive force appeared again, tending asymptotically to the value characteristic of the half-space (in this case, it was greater than at zero thickness, since the layer material was more compliant than the half-space material). It is interesting to note at a certain layer thickness, the positive force was greater than that observed in the case of a half-space.

Another important input parameter is the relative compliance of the layer (Figure 11). We chose here the thickness of the layer, which provided the separation effect for the relatively compliant layer. In this case, the deformation of the layer was much greater than that of the substrate. With an increase of the layer elastic modulus, the contact zones decreased and became relatively farther from each other. Mutual effect and tangential forces became weaker. The deformation of the substrate became comparable to the deformation of the layer, which led to the appearance of a positive tangential force. There is a value of χ at which the positive force reached its maximum value, then decreased.

It is possible to compare the results shown in Figure 12 with the experimental results. Figure 12 shows the dependence of the arising tangential forces on the distance in the cases of a relatively thin (dashed line) and a relatively thick (solid line) layers. When the layer is thin, the bulging of the material, leading to the appearance of negative tangential forces, occurs near the contact area and decreases sharply with the distance from this area. The thicker the layer, the further from the contact area is the zone of bulging. This explains the fact that when the balls move apart on a relatively thick layer, the positive tangential force is replaced by a negative one. In the case of a thin layer, the opposite phenomenon occurs due to the fact that, at a sufficiently large distance, the mutual effect associated with the deformation of the layer becomes negligible, but the mutual effect provided by the deformation of the substrate remains. In our case, the substrate effect was negligible, because its material was much more rigid and more compressible than the material of the layer. Therefore, the initial position of the balls can determine whether separation or convergence of the balls will occur. This also explains the experimental fact that the balls stopped after separation in a position of negligible tangential force.

## 4. Conclusions

An experimental method was developed to demonstrate the effects of convergence or separation of particles in the intermediate layer between two bodies during friction. Steel balls were used as the physical model of the particles. The convergence effect was obtained for a relatively thick compliant layer in contact with the balls, which had different initial configuration. Some irregularities in the configuration were also considered as a starting condition and changed the convergence process. The separation effect was obtained for two balls in contact with a relatively thin layer of a low compressible elastic material. An analytical–numerical model based on double Fourier transforms, boundary element method, and iterative procedure was developed to study tangential forces arising due to the mutual effect related to the contact of two balls with a two-layered half-space. We found that the sign of the tangential force depended on the layer thickness, its relative compliance, and the distance between the balls. The experimental observation of a fixed position of the balls after their separation could be explained: this position corresponded to a negligible tangential force, as shown in Figure 12.

It should be noted that tangential forces arising due to the mutual effect do not affect the friction force at the macro level. (If we add up the forces acting on all the balls, we obtain zero). However, the assembly or separation of particles can significantly affect the adhesion or tribochemical processes and, thus, the friction force.

These results open up interesting prospects. Further research in this direction may be interesting from the point of view of creating structures and controlling motion in the intermediate layer between two bodies.

## Figures and Tables

**Figure 1 materials-14-05689-f001:**
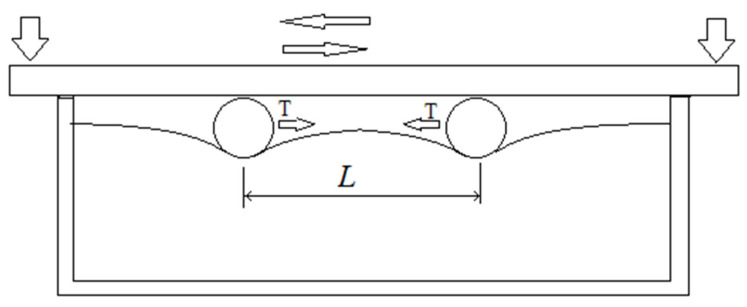
Scheme of the experiment.

**Figure 2 materials-14-05689-f002:**
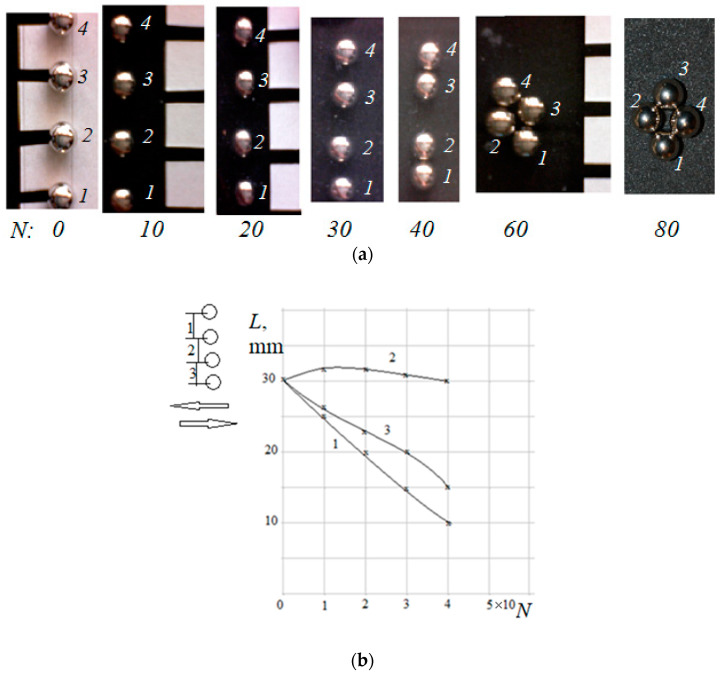
A line of four equidistant balls. The initial distance between the centers in the line was 25 mm. Movement along the line of balls (**a**). Change of the distance between the centers (**b**). *N* is the number of cycles.

**Figure 3 materials-14-05689-f003:**
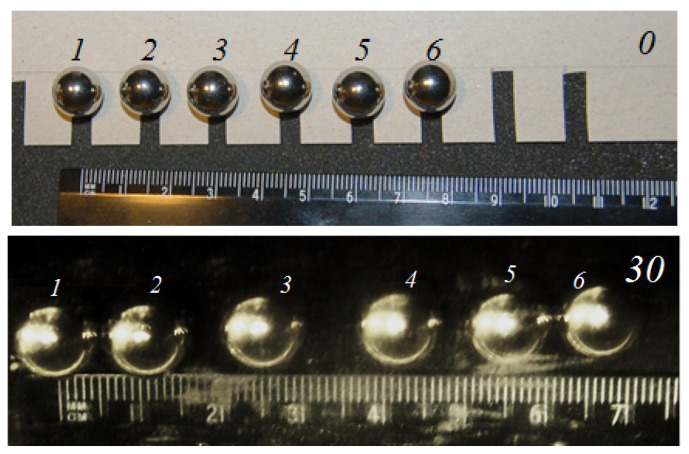
Movement along the line of balls (the initial distance between the centers in the line was 15 mm). The number of cycles is reported in the upper right corner.

**Figure 4 materials-14-05689-f004:**
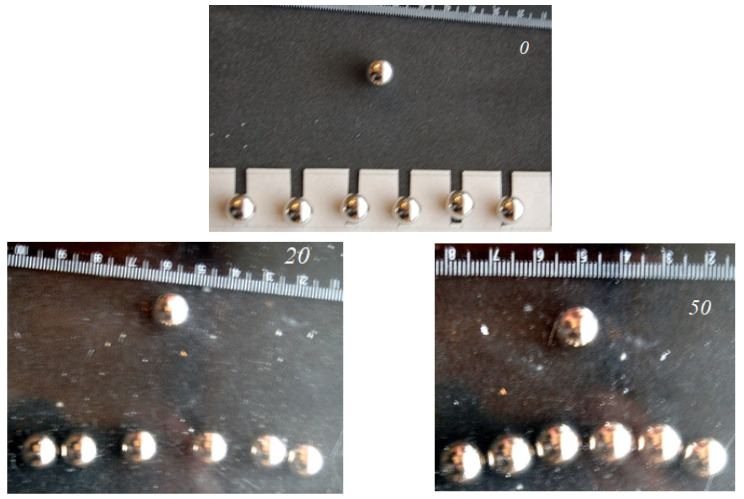
Reciprocating motion along the line of balls (the initial distance between the centers in the line was 15 mm). The number of cycles is reported in the upper right corner.

**Figure 5 materials-14-05689-f005:**
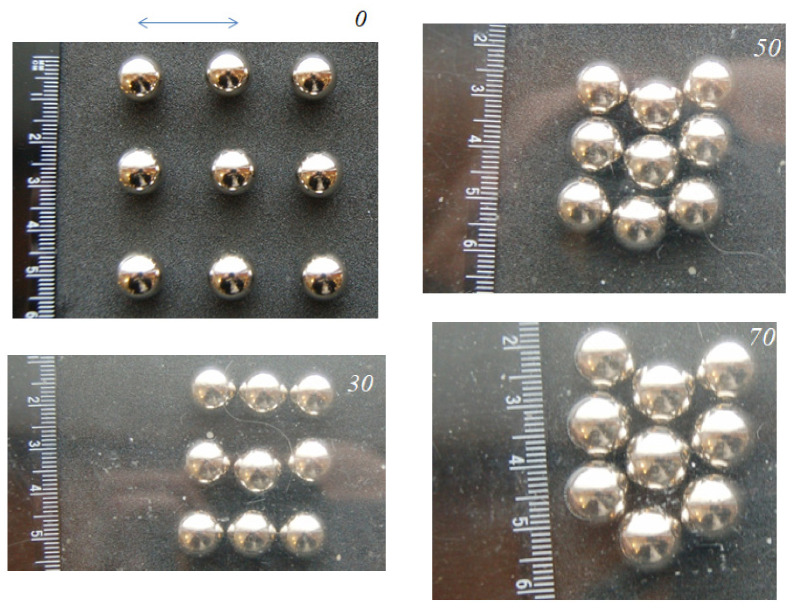
Nine balls at an initial distance of 20 mm from each other.

**Figure 6 materials-14-05689-f006:**
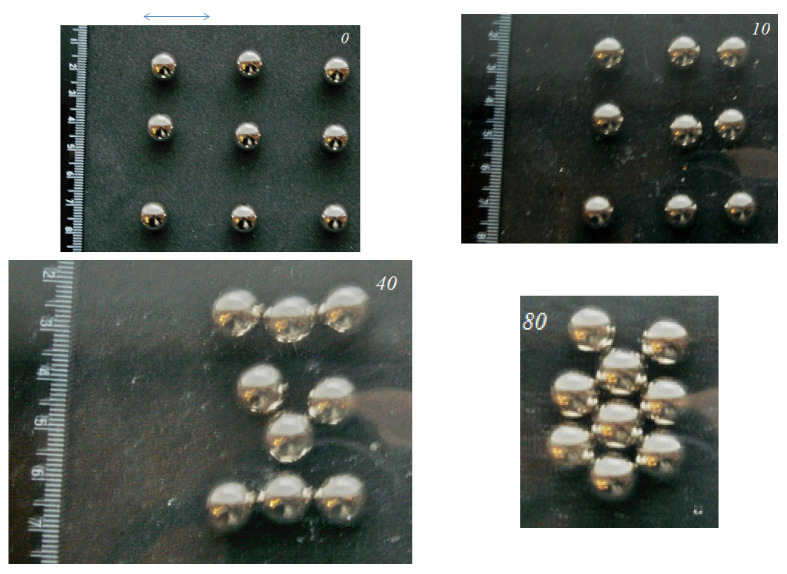
Nine balls at an initial distance of 30 mm from each other.

**Figure 7 materials-14-05689-f007:**
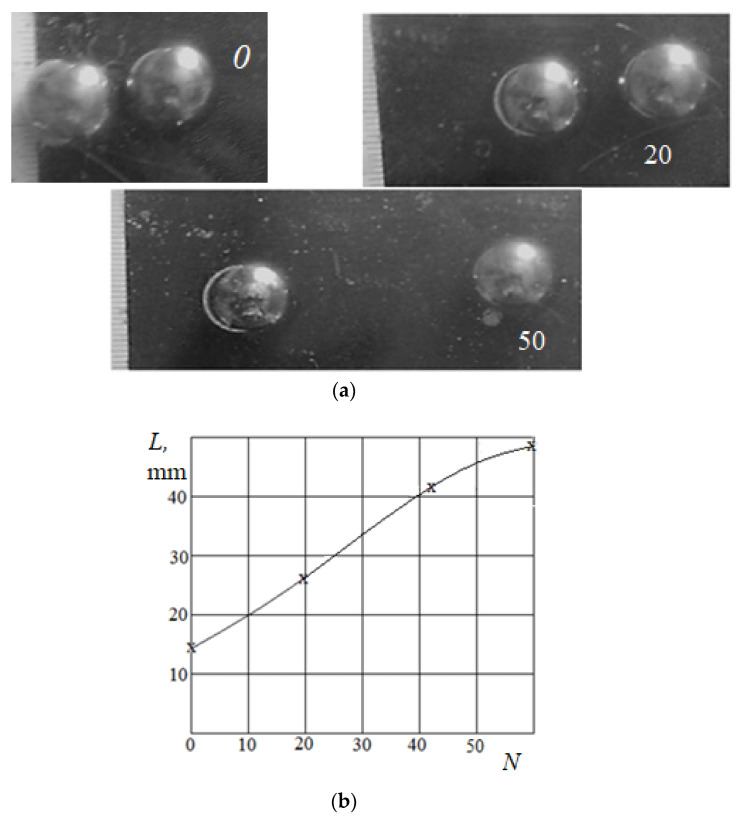
Effect of ball separation (**a**), evolution of the distance between the ball centers (**b**). *N* is the number of cycles.

**Figure 8 materials-14-05689-f008:**
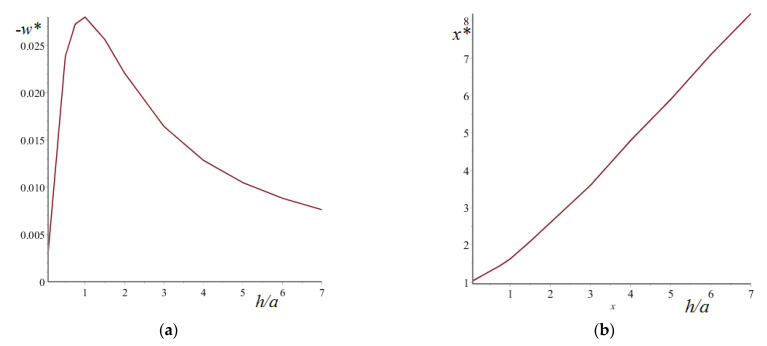
Dependences of $w^{*}$ (**a**) and $x^{*}$ (**b**) on the thickness of the compliant layer. $E_{1}/E_{2}=10^{-3}$, $\nu_{1}=0.48$, $\nu_{2}=0.3$, $w^{*}=\frac{w}{a}\;\frac{E_{1}}{q}$, $x^{*}=x/a$.

**Figure 9 materials-14-05689-f009:**
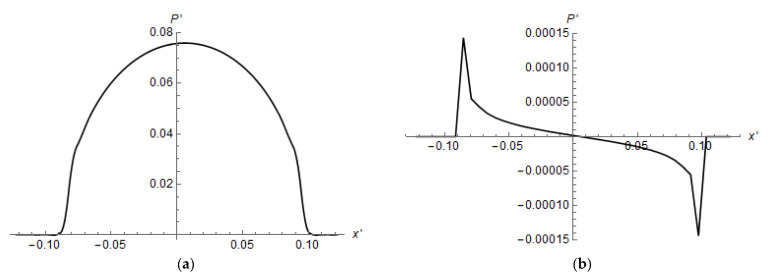
Contact pressure distributions (**a**), difference between contact pressures under left and right balls (**b**). $E_{1}/E_{2}=10^{-3}$, $\nu_{1}=0.48$, $\nu_{2}=0.3$, $L^{\prime }=2.0$, $h^{\prime }=1.16$, $Q^{\prime }=2.66(6)\times 10^{-3}$.

**Figure 10 materials-14-05689-f010:**
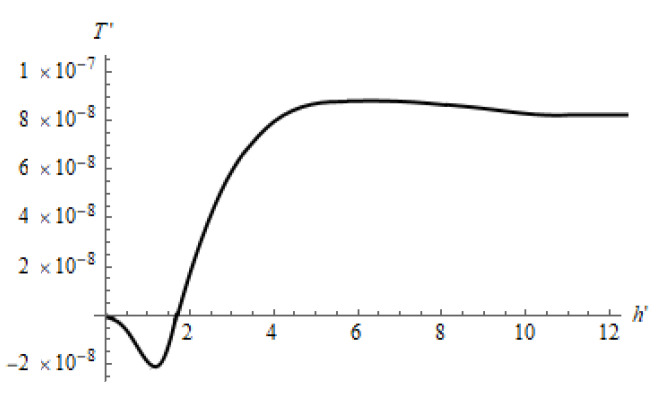
Tangential force as a function of the layer thickness. $E_{1}/E_{2}=10^{-3}$, $\nu_{1}=0.48$, $\nu_{2}=0.3$, $L^{\prime }=2.0$, $Q^{\prime }=2.66(6)\times 10^{-3}$.

**Figure 11 materials-14-05689-f011:**
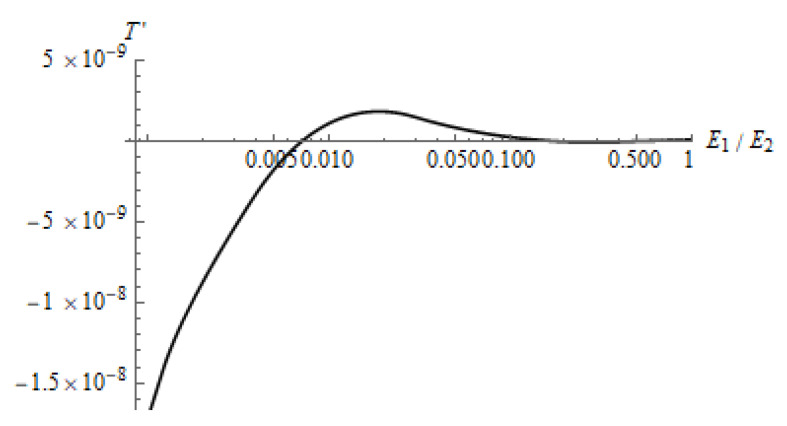
Dependence of the tangential force on the elastic modulus of the layer. $\nu_{1}=0.48$, $\nu_{2}=0.3$, $L^{\prime }=2.0$, $Q^{\prime \prime }=Q/\left(60\times 10^{6}\times R^{2}\right)=2.66(6)\times 10^{-3}$.

**Figure 12 materials-14-05689-f012:**
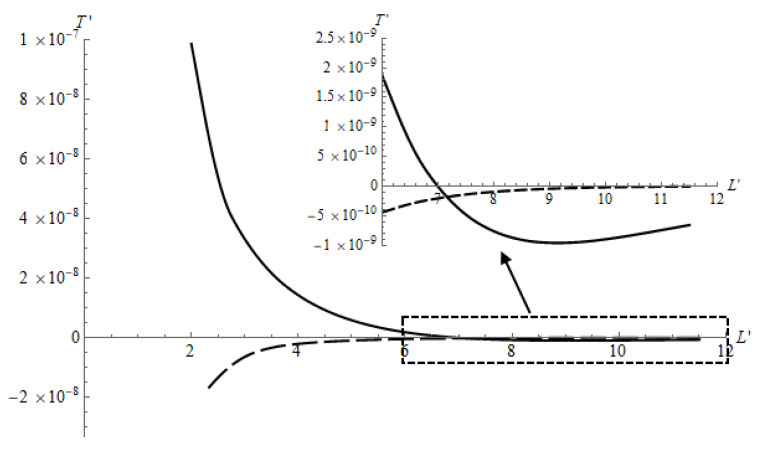
Tangential force as a function of the distance between the ball centers. $E_{1}/E_{2}=10^{-3}$, $\nu_{1}=0.48$, $\nu_{2}=0.3$, $h^{\prime }=1.16\;(\mathrm{dashed}\;\mathrm{line})$, $h^{\prime }=6.0\;(\mathrm{solid}\;\mathrm{line})$, $Q^{\prime }=2.66(6)\times 10^{-3}$.

## Data Availability

Not applicable.

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
