# Peer review of "Theoretical and Experimental Study of Changes in the Structure of the Intermediate Layer during Friction between Contacting Bodies"

_materials, 2021, doi:10.3390/ma14195689_

Round 1

Reviewer 1 Report

In this work, the authors studied the role of intermediate particle layer on the sliding forces using both experimental and numerical approaches. The topic is interesting and comply with the scope of this journal. However, the overall quality of this manuscript can be further improved, particularly some of the expressions need to be clarified. It can be published if the following comments and questions are well addressed.

I didn’t see any comparisons between numerical calculations and experimental measurements?

If the vertical displacement was controlled, normal loads can be different among the cases studied.

How to understand effects from the edge as an artifact?

Is the numerical model able to address multi-ball system?

How the contact area was determined?

Is there any evidence on the ‘bulging’ claimed in the text so that the forces are negative at small separations in the case of thin layer?

Objects in the figures should be clearly labeled. Figure captions should be informative.

Author Response

The authors are grateful for reviewing the paper. Questions and answers are below.

1. I didn’t see any comparisons between numerical calculations and experimental measurements?

We could use our model to do the calculations exactly for the experimental conditions. But in this case, the results would be local. We obtained a qualitative agreement with the experiment (the last paragraph of Section 3.3 and Conclusions, which are revised), and also performed a complete analysis depending on the input parameters. Such an analysis with including the limiting cases (half-spaces with the elastic properties of the layer and the substrate for zero and infinite layer thickness) also makes it possible to verify the model.

2. If the vertical displacement was controlled, normal loads can be different among the cases studied. How to understand effects from the edge as an artifact?

This is certainly true. Thanks for this comment. We have highlighted the effect of edges in the analysis of the experimental results:

Such type of grouping is caused by the ‘edge effect’: it means that, having the same penetration, the edge balls are loaded greater than the central ones [1]. Two groups are formed and then interacted with each other. (Page 4)

3. Is the numerical model able to address multi-ball system?

Initially, the model was developed for a system of an arbitrary number of indenters. Since the analysis was carried out for two balls, the problem was also formulated for two balls. As per your comment, we have substantially revised Sections 3.1 and 3.2 to offer the multiple contact problem. In the second paragraph of Section 3.3, we have added text on the motivation for choosing a system of two balls for analysis:

We chose a two-ball configuration for the analysis of tangential forces due to mutual influence, since the separation was experimentally obtained for this configuration. In addition, having two equally loaded balls, we can reduce a large number of input parameters of the problem; this is convenient for analysis. (Page 13)

4. How the contact area was determined?

In the new version of Section 3.2, the answer is as follows:

According to the boundary conditions (1), (3) the unknown contact pressure should be 0 outside the contact area. The systems (10) and (11) do not take into account this fact, so a solution will include negative pressure elements. In order to obtain the unknown contact areas, one should consider these elements to be zero and reduce the system rank due to this. After, the reduced system is solved again. The described iteration procedure con-tinues until there are no negative pressure elements in the solution. As result one obtain the unknown contact areas, contact pressure and the normal displacement of indenters. (Page 12)

5. Is there any evidence on the ‘bulging’ claimed in the text so that the forces are negative at small separations in the case of thin layer?

We have added a new Figure 8 and the corresponding paragraph at the beginning of Section 3.3 to describe and analyze the bulge effect (page 13).

6. Objects in the figures should be clearly labeled. Figure captions should be informative.

In the photographs, the balls were numbered where their mutual movements are not always obvious (Figures 2 and 3). Also, the caption to Figure 9 (former Figure 8) has been corrected.

We had only 10 days for corrections and tried to do our best.

Reviewer 2 Report

Reasoning

  1. The problem is posed to change the position of spherical deformable bodies on an elastic surface under the action of small perturbing tangential forces. Usually this problem is solved in a different way, that is, there is a position in which the minimum deformation stresses are achieved. It is therefore clear that for a thick elastic substrate, it is more preferable that the balls form one flat body (grouped together).  For a thin elastic plate, the minimum curvature must be sought. With a sufficient distance of the balls from the fixed edges of the plate or with loose edges (as in this case), the minimum curvature can be achieved with dispersed deforming bodies.
    As noted in the article, depending on the properties of the elastic substrate, the forming forces can have a different sign (drawing together or separating).
    In any case, the presented experiment cannot be considered a friction contact simulation containing rolling bodies. Since the deformation stresses of the surfaces are not modeled, the relationship between the elasticity of the rolling elements and the surface is not taken into account. The described experiment is an imitation, not a simulation of the behavior of rolling bodies in a friction contact. The problem is interesting from the point of view of mechanics.

Comment

  1. The quality of the submitted photos needs to be improved. Modern photography equipment allows you to get clear images.

Author Response

The authors are very grateful for careful reading of the paper and positive feedback.

 Unfortunately, all photos (except the initial configurations) were taken through glass. This was necessary for the purity of the experiment, but deteriorated the quality of the images. The camera inside the system did not allow us to see the big picture. We noted the fact of photographing through glass in the first paragraph of Section 2. In addition, we added sharper images in Figure 3 (30 cycles) and Figure 9 (80 cycles).

Reviewer 3 Report

The paper contains a high quality of scientific work but please solve the following issues:

  1. In Introduction the first 4 lines are identical with the first 4 lines from Abstract. Please, try to re-formulate the text.
  2. The same problem in lines 84, 85, 86, 87 with the text from abstract.
  3. Please indicate, by using numbers or names, the signifiance of the parts.
  4. Line 112: please explain the meaning of E=57.4 kPa – you claim that is the effective stiffness of the rubber. The stiffness is measured, generally, in N/m.
  5. How was established the Poisson’s ratio from line 173?
  6. It would be nice to start with a scketch for the Problem formulation.
  7. Please explain how the theoretical model supports the experiments.
  8. How the experimental and theoretical results fits into the specific literature? Please find more references to explain that.

Author Response

The authors are very grateful for careful reading, comments and positive feedback. Questions and answers are below.

1. In Introduction the first 4 lines are identical with the first 4 lines from Abstract. Please, try to re-formulate the text. The same problem in lines 84, 85, 86, 87 with the text from abstract.

We corrected the text in Introduction.

2. Please indicate, by using numbers or names, the signifiance of the parts.

Sorry, we did not understand this comment.

3. Line 112: please explain the meaning of E=57.4 kPa – you claim that is the effective stiffness of the rubber. The stiffness is measured, generally, in N/m.

Many thanks. We replace “compliance” with “elastic modulus”.

4. How was established the Poisson’s ratio from line 173?

This value is typical for rubbers. We added this note to the text.

5. It would be nice to start with a scketch for the Problem formulation.

We add the starting paragraph in Section 3:

The study of mutual effect in the problem of multiple contacts is usually associated with the surface roughness. For the case of elastic half-space the periodic system of dies as well as fixed number of dies were considered in [1]. Mutual effect leads to not uniform force distribution between a limited assembly of dies with controlled penetration, as well as to arising of tangential force because of non-symmetric pressure distribution. Tangential force of such type was analyzed for the case of a single slider in contact with elastic [17] and viscoelastic [18] solids. For the case of asperities such tangential force cannot give any significant effect. The contact problem for the system of balls is mathematically identical to the problem for the system of dies, but the balls can change their location due to the tangential force.

6. Please explain how the theoretical model supports the experiments.

We could use our model to do the calculations exactly for the experimental conditions. But in this case, the results would be local. We obtained a qualitative agreement with the experiment (the last paragraph of Section 3.3 and Conclusions, which are revised), and also performed a complete analysis depending on the input parameters. Such an analysis with including the limiting cases (half-spaces with the elastic properties of the layer and the substrate for zero and infinite layer thickness) also makes it possible to verify the model.

7. How the experimental and theoretical results fits into the specific literature? Please find more references to explain that.

This research is not in the mainstream, so it is extremely difficult to find relevant literature. The effect of mutual influence was usually investigated in the sight on roughness, and not on particles. Nevertheless, we have added two publications containing examples of the occurrence of shear forces due to asymmetric pressure distribution. (First paragraph of Section 3).

Reviewer 4 Report

The paper presents self organization of particles in a tribo contact in a novel and highly interesting approach, recommended for further research. The mathematical description, even though not quite understandable gives a reasonable explanation for the experimental data. The paper is thus recommended for publication.  However some weaknesses are apparent and have to be taken out :Within the introduction, same as in the abstract, the motivation of the study is not clear :

a) is it, to explain friction in a contact and if so : how is the friction coefficient related to the findings ?

b)  is it, to explain dry lubrication in general or are coatings in mind ? 

c) is the intention to come to a model for prediction low/high friction in a dry tribocontact and what are the predictors ? This could be of highest interest  in the field of dry clutches !

d) if it is not coating, instead the general behaviour of particles in a dry lubricated contact, a DoE study, comprising different and randomly sized and arranged particles and different Poisson ratio is missing - there is a hint at the end of the paper, that this study will go toward such a goal, but still the link is not given to any aspect in reality.

e) other models, e.g. Ostwald ripening, chemical nature of dies are not mentioned.

I'll give the advice to improve the introduction significantly, such, that the reader is aware of the intention, e.g. to predict friction in whatever technical application (f.e. dry clutches) and to tell the reader whether the intention is, to predict friction in  dry lubricated contacts or to predict friction by the use of coatings and also to clearly separate the paper from lubricated contact. 

Nevertheless the paper appears very interesting in bringing up a novel approach to the community. 

Author Response

The authors are very grateful for positive feedback and interesting ideas: we received the good idea in improving the motivation part. Questions and answers are below.

a) is it, to explain friction in a contact and if so : how is the friction coefficient related to the findings ?

In this study, only the mechanical component of the interaction of the particles of the third body is investigated. The effect of energy loss on the collision of particles in their closed position is not considered. This effect leads to increased friction losses. Microscale also assumes adhesion of the particle to the contact surfaces [14]; when approaching, molecular interaction between the particles also occurs [15], which additionally increases the dissipation of energy during friction. Probably, the assembly or dispersal of particles should affect tribochemical processes, for example, oxidation [16]. The real third body consists of microparticles of various sizes, particles can be of different nature, shape and have different mechanical properties. The physical and mathematical models presented here make it possible to isolate the effect of mutual influence from a large number of factors influencing the behavior of the particles of the third body. (Page 3, new text in the Introduction)

It should be noted that tangential forces arising due to the mutual effect do not affect the friction force at the macro level. (If we add up the forces acting on all the balls, we get zero.) But the assembly or separation of particles can significantly affect the adhesion or tribochemical processes and, thus, the friction force. (Page 16, new text in Conclusions)

b) is it, to explain dry lubrication in general or are coatings in mind ?

Both experimental  and theoretical study can be used for coatings and a bulk materials (very thick layers), but separating effect occurs only for compliant coatings.

c) is the intention to come to a model for prediction low/high friction in a dry tribocontact and what are the predictors ? This could be of highest interest in the field of dry clutches !

To predict friction, we should use our results and combine them with models of adhesion (or thermal effects or chemical changes…).

d) if it is not coating, instead the general behavior of particles in a dry lubricated contact, a DoE study, comprising different and randomly sized and arranged particles and different Poisson ratio is missing - there is a hint at the end of the paper, that this study will go toward such a goal, but still the link is not given to any aspect in reality.

e) other models, e.g. Ostwald ripening, chemical nature of dies are not mentioned.

We did not set out to write about all the processes taking place in the third body. It is large interesting topic. We mentioned them in a new paragraph in the introduction with the appropriate references (as examples).  Paragraph 3 from the Introduction also includes some references. We believe that the mechanical component of particle interaction always takes place.

Round 2

Reviewer 1 Report

The revised manuscript is improved and the authors well explained the questions. I thus have no further comments and agree to have this work published.